# RANK++LETR: Learn to Rank and Optimize Candidates for Line Segment Detection

**Xin Tong**     **Baojie Tian**     **Yufei Guo**     **Zhe Ma**[*]
Intelligent Science & Technology Academy of CASIC
xin_tong@pku.edu.cn, mazhe_thu@163.com

## Abstract

It is observed that the confidence score may fail to reflect the predicting quality accurately in previous proposal-based line segment detection methods, since the scores and the line locations are predicted simultaneously. We find that the line segment detection performance can be further improved by learning-based line candidate ranking and optimizing strategy. To this end, we build a novel end-to-end line detecting model named RANK++LETR upon deformable DETR architecture, where the encoder is used to select the line candidates while the decoder is applied to rank and optimize these candidates. We design line-aware deformable attention (LADA) module in which attention positions are distributed in a long narrow area and can align well with the elongated geometry of line segments. Moreover, we innovatively apply ranking-based supervision in line segment detection task with the design of contiguous labels according to the detection quality. Experimental results demonstrate that our method outperforms previous SOTA methods in prediction accuracy and gets faster inferring speed than other Transformer-based methods.

## 1 Introduction

Line segments and junctions are crucial information in structured scenes and are ubiquitous in human-made environments. An accurate line segment detection algorithm can significantly enhance various computer vision applications, such as 3D reconstruction [36; 2], camera calibration [22; 27], depth estimation [40], scene understanding [11], object detection [29], SLAM [18; 41], etc. Traditional geometric-based line segment detection algorithms usually extract low-level image features and group them into line segments. These methods often run at a fast speed, while may suffer from fragmented prediction. Learning-based methods achieve promising results by learning knowledge from image sets with supervision, which are able to detect longer and more meaningful line segments.

Proposal-based methods constitute a pivotal component within learning-based approaches and have been extensively studied recently. These models typically output target predictions such as endpoint coordinates or midpoint coordinates with endpoint offsets for line segments. Generally, these methods first simultaneously predict both the positional coordinates and confidence scores of candidate line segments, then select the top-ranked proposal-based on confidence scores as final predictions. Previous study [31] points out that some accurately detected line segments are assigned low confidence scores during prediction since confidence prediction and location regression of line segments are independent. Specifically, given the simultaneously predicted line candidates with confidence scores and positions, the detection performance can be significantly improved even if only proper scores are assigned. Based on the observation, we find that the line segment detection performance can be further improved by learning-based line candidate ranking and optimizing strategy.

Transformers depend on attention modules to gather relevant features. However, in the classic deformable attention module, the attention position of a query is usually around a reference point on

---

[*]Corresponding author

the feature map, which is not easy to adapt to the long and narrow area for detecting line segments. Thus, we specially design a novel attention module named line-aware deformable attention (LADA), which can align well with the elongated geometry of line segments for better perceiving line features.

To effectively train our model for quality-aware ranking of line candidates, proper supervision is essential. Ranking-based losses aim to rank the positive predictions above negative ones and sort the high-score candidates over low-score ones, which naturally suit our line ranking task. We define the contiguous label according to the quality of the predictions, which are based on the distance of the nearest ground truth and the predictions. Then, ranking-based losses can be applied to promote higher scores for high-quality predictions.

In this work, we propose a novel end-to-end line detecting model named RANK++LETR upon deformable DETR architecture. For Transformer-based line detection methods, LETR uses DETR architecture where the backbone and the encoder are used to extract features and the decoder is used to generate line segments. RANK-LETR adopts an encoder-only network and directly predicts the lines from the encoder. Different from the above approaches, our method leverages the complete network architecture of deformable DETR in design philosophy. Specifically, we apply distinct supervision during the encoder and decoder stages: the encoder is guided to predict candidate line segments, while the decoder is responsible for ranking and refining these candidate line segments.

Our contributions can be summarized as follows: (1) We present a novel DETR-like line segment detection framework, where the encoder is used to generate candidate line segment proposals, while the decoder is used to optimize their confidence scores and locations. (2) We propose line-aware deformable attention where the perception field can be long and narrow to catch features along candidate line segments. (3) By defining the continuous label for line segment detection, we employ ranking-based supervision to optimize confidence scores in the decoder. (4) Experimental results demonstrate that our method outperforms previous SOTA approaches in prediction accuracy while running faster than previous Transformer-based models.

## 2 Related Work

### 2.1 Line Segment Detection.

Traditional line detection methods often rely on grouping image gradient [32; 1; 21] and pre-defined rules [9; 7; 34]. Recently, learning-based methods have achieved promising results. For junction based methods, DWP [13] predicts junction map and edge map in two branches before merging them. PPGNet [43] uses a point-pair graph to describe junctions and line segments. L-CNN [45] applies line proposal and LoI pooling to propose candidate lines and verify them. Methods with dense prediction first predict representation map and extract line segments with post-processing. AFM [35] proposes attraction field maps to represent the image space and uses a squeeze module to generate line segment maps. HAWP [38] builds a hybrid model considering 4D attraction field and further extends to holistic attraction field [37]. Lin *et al.* [17] apply deep Hough transform to the previous detection architectures. TP-LSD [14] introduces tri-points line segment representation for end-to-end detection. M-LSD [8] presents SoL augmentation and designs an extremely efficient architecture for fast detection. SOLD2 [26] and DeepLSD [25] apply unsupervised pipelines and are able to detect fine and sufficient line segments. Transformer-based method can directly output the locations of the line segments. LETR [33] models it as object detection and predicts line segments with DETR architecture. RANK-LETR [31] applies match predicting and re-ranking to improve the training efficiency and the recall of high quality predictions. In this work, we extend Transformer and proposal-based method with a pure learnable optimization module for better performance.

### 2.2 Visual Transformer for Detection

Visual Transformer for object detection task is originated in DETR [3], in which a Transformer-based encoder-decoder framework is adopted and end-to-end supervision is applied with bipartite matching. Zhu *et al.* [46] further proposes deformable DETR in which each query only focuses on a small set of keys with learnable locations. A denoising training method is presented in [16] and a query formulation using dynamic anchor boxes is introduced in [20] to speed up training convergence of DETRs, which are further extended in DINO [42] Hou *et al.* [12] designed a hierarchical query filtering strategy to reduce the computational redundancy of DETR. Visual Transformer is widely

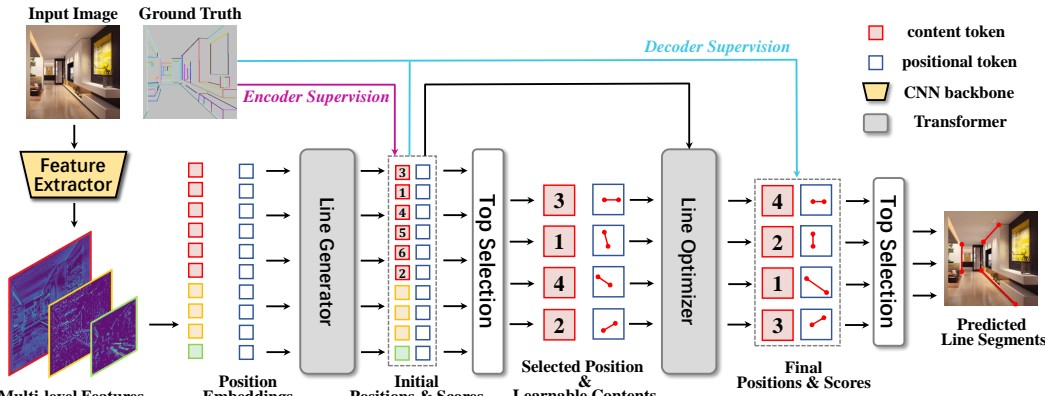

Figure 1: Overview of the proposed RANK++LETR. An image is first fed into a CNN-based feature extractor and multi-level feature maps are obtained from different layers. Then, initial line segments are generated by Transformer encoder based line segment generator with the multi-level features. Candidate line segments with high confidence scores are selected to initialize the reference points in the proposed line-aware deformable attention of the line optimizer. Finally, the confidence scores and positions of the candidate line segments are optimized, and we choose the candidate line segments according to their optimized confidence scores with non-maximum suppression as our final prediction. The entire method can be end-to-end training and inferring.

used in many other visual detection tasks. Xu *et al.* [33] apply DETR in line segment detection with a multi-scale encoder-decoder strategy. Tong *et al.* [30] use Transformer decoders in end-to-end vanishing point detection with Gaussian hemisphere division. Chenhang *et al.* [10] apply Transformer in 3D object detection with a set-to-set translation strategy. Tan *et al.* [28] utilize Transformers to represent context features and line segments for detecting and reconstructing 3D planes from a single image. Liu *et al.* [19] employed a Transformer-based architecture for end-to-end lane detection. Leveraging the encoder-decoder architecture of deformable DETR, we design to generate line segment proposals and optimize line candidates successively in a single framework.

### 2.3 Ranking-based Losses

Ranking-based losses have received much attention in recent studies. Chen *et al.* [5] first propose Average Precision Loss to address the imbalance of foreground-background classification problem by framing object detection as a ranking task. Rank & Sort (RS) Loss that defines a ranking objective between positives and negatives as well as a sorting objective to prioritize positives with respect to their continuous IoUs is designed in [24]. Yavuz *et al.* [39] apply Bucketed Ranking-based (BR) Losses which group negative predictions into several buckets. Cetinkaya *et al.* [4] extend ranking-based Loss to edge detection with uncertainty modeling. Ranking-based loss is also used in 3D reasoning frameworks [15]. In this work, we naturally employ ranking-based losses to supervise the line optimizer for better confidence prediction.

## 3 Method

### 3.1 Line Segment Detection Modeling

Building upon deformable DETR encoder-decoder architecture, we model line segment detection as an end-to-end process including line proposal generation and optimization, with each line proposal parameterized by the position of its endpoints and the confidence score. Specifically, the encoder generates candidate line segment proposals with associated confidence scores and positions, while the decoder performs optimization through positional refinement and confidence re-ranking. The entire pipeline can be end-to-end training and inferring.

As shown in Fig.1, the proposed method takes images as input and finally predicts a given number of line segments sorted according to their confidence. It is initiated by processing input images through a CNN backbone to capture multi-level feature representations. Then these features subsequently

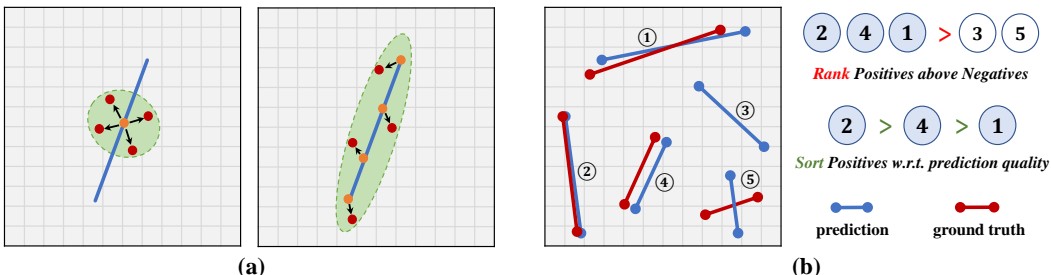

Figure 2: (a) The attention positions (red points) of a query are usually around a reference point (orange point) on the feature map in classic deformable attention module (left). In our proposed line-aware deformable attention (LADA, right), the attention positions are distributed in a long and narrow area and align well with the elongated geometry of candidate line segments (blue line). (b) Ranking-based supervisions rank the positives above the negatives and sort the positives with respect to their prediction qualities. We define the prediction quality relating to the distance between a predicted line segment and its nearest ground truth. The positives and negatives are also distinguished according to prediction quality, e.g., with a proper threshold.

undergo hierarchical encoding via a deformable Transformer encoder for candidate line proposal generation. The encoder implements a spatial-aware assignment mechanism where each spatial feature point is assigned to detect line segments whose centroid resides within its proximal receptive field. According to the confidence score predicted by the encoder, line segments with high scores are chosen as candidates for the decoder. In this work, we rank line segments with their prediction quality and refine their location simultaneously. The decoder takes learnable content and candidate line segment positions as initiating and gradually gathers information from the encoder feature maps. Finally, the decoder predicts the optimized confidence scores and their locations simultaneously in a pure learning-based manner.

## 3.2 Line-Aware Deformable Attention Module

In DETRs, each query adaptively aggregates information from feature vectors at specific attention positions in the corresponding feature map with the attention module. Typically, given the query vector $z_q$ and feature map $x$, the multi-head attention module can be represented as

$$\mathcal{A}(\boldsymbol{z}_q, \boldsymbol{x}) = \sum_{h=1}^{H} \boldsymbol{W}_h [\sum_{k=1}^{K} A_{hqk} \cdot \boldsymbol{W}'_h \boldsymbol{x}(\boldsymbol{p})], \tag{1}$$

where $h$ indexes the attention head and $k$ indexes the sampled keys. $\boldsymbol{W}_h$ and $\boldsymbol{W}'_h$ are learnable projection weights. $A_{hqk}$ is the *Scaled Dot-Product Attention* weight. $\boldsymbol{p}$ indicates the attention position of query vector $\boldsymbol{z}_q$ on feature map $\boldsymbol{x}$ at the $h$-th head and $k$-th sampled point. Here we adopt similar notations as deformable DETR [46] for a better understanding.

For the classical multi-head attention module, the attention positions are generally predefined to cover all feature point locations across the entire feature map. Thus, $\boldsymbol{p}$ can be defined as the $k$-th index of the predefined locations $\mathbf{G}$

$$\boldsymbol{p} = \mathbf{G}[k]. \tag{2}$$

Deformable attention introduces learnable deformable attention mechanism, where the attention positions for each query are determined by a reference point $\boldsymbol{p}_q$ and multiple learnable offsets, which can be determined as

$$\boldsymbol{p} = \boldsymbol{p}_q + \mathcal{F}_{hk}(\boldsymbol{z}_q), \tag{3}$$

where $\mathcal{F}_{hk}(\boldsymbol{z}_q)$ is a function of $\boldsymbol{z}_q$, e.g., a linear function. It allows the attention positions to be dynamically adapted based on the query.

Since our method uses attention module for line segment verification and refinement, the query vector should focus on regions near the corresponding candidate line segment. These regions are generally long and narrow as shown in Fig.2 (a). However, in the classic deformable attention module, the attention position of a query is usually around a reference point on the feature map, which is not

easy to adapt to the long and narrow area for detecting line segments. To address this limitation, we propose a line-aware deformable attention (LADA) module, where the attention positions are aligned with the elongated geometry of candidate line segments. Specifically, $p$ in LADA module can be represent as

$$p = \frac{k-1}{K-1} s_q + \frac{K-k}{K-1} e_q + \mathcal{F}_{hk}(z_q), \tag{4}$$

where $s_q$ and $e_q$ are two endpoints of the candidate line segment $l_q$. With this design, the receptive field of the attention module is distributed along the candidate line segments. It enables the model to better perceive their alignment with semantically image features, e.g., edges and endpoints.

### 3.3 Ranking Line Candidates with Prediction Quality

For further optimizing the ranking through the prediction quality of the candidate line proposal, we employ ranking-based losses to supervise the confidence scores. The key to applying ranking-based supervision is to define contiguous labels that can reflect the line detection quality properly. We define a simple yet efficient contiguous label $l_i$ based on the distances between the predicted lines and the nearest ground truths of them. Specifically, it can be defined as

$$l_i = max(0, 1 - \delta_l * min(\|e_i - e_i^*\|_2 + \|s_i - s_i^*\|_2, \|e_i - s_i^*\|_2 + \|s_i - e_i^*\|_2)), \tag{5}$$

where $e_i, s_i$ means the two endpoints of the $i$-th line segment and $e_i^*, s_i^*$ are two endpoints of the corresponding ground truth. $\delta_l$ is a factor that controls the threshold of distance. The ranking-based solution we used consists of two components that are visually exhibited in Fig.2 (b) for better understanding.

*Ranking Positives over Negatives.* Given the candidate line segments and their contiguous labels, ranking loss is used to rank the positives above the negatives. We consider the lines whose $l_i > 0$ as positives while others are considered as negatives. $\mathbf{P}$ indicates the set of positive line segments and $\mathbf{N}$ indicates the set of negatives. We define the ranking loss $\mathcal{L}_{rank}$ using a differentiable approximation of Average Precision following [4], which can be presented as

$$\mathcal{L}_{rank} = 1 - \frac{1}{|\mathbf{P}|} \sum_{i \in \mathbf{P}} \frac{\sum_{j \in \mathbf{N}} H(x_{ij})}{\sum_{j \in \mathbf{P} \cup \mathbf{N}} H(x_{ij})}, \tag{6}$$

where $H(x_{ij}) = max(1, min(0, (c_j - c_i)/2\delta_H + 0.5))$ is the step function with a $\delta_H$-approximation around the step. $c$ is the confidence score.

*Sorting Positives with Prediction Quality.* Each candidate has a different prediction quality according to the alignment degree with the corresponding ground truth, i.e., $l_i$. Thus, we supervise to sort the positive line segments with $l_i$, making the well-aligned predictions tend to get higher confidence scores. To this end, we use the sorting objective $\mathcal{L}_{sort}$ introduced in [24], which can be presented as

$$\mathcal{L}_{sort} = \frac{1}{|\mathbf{P}|} \sum_{i \in \mathbf{P}} \left( \frac{\sum_{j \in \mathbf{P}} H(x_{ij})(1 - l_i)}{\sum_{j \in \mathbf{P}} H(x_{ij})} - \frac{\sum_{j \in \mathbf{P}} H(x_{ij})[l_j \geq l_i](1 - l_j)}{\sum_{j \in \mathbf{P}} H(x_{ij})[l_j \geq l_i]} \right), \tag{7}$$

where the former term is the current sorting error and the latter term is the target sorting error, respectively. More details are referred to [24].

### 3.4 Training Strategy

During training, supervisions are added on both Transformer encoder and decoder, called encoder supervision and decoder Supervision, respectively. For the encoder supervision, binary cross-entropy loss $\mathcal{L}_{conf}^{E}$ and L2 loss $\mathcal{L}_{pos}^{E}$ are applied to supervise the confidence scores and positions of the predicted line segments. For the decoder Supervision, we use ranking-based supervision $\mathcal{L}_{rank}$ and $\mathcal{L}_{sort}$ to learn ranking the candidate line segments. Moreover, L2 loss $\mathcal{L}_{pos}^{D}$ is also used to optimize the positions of line segment candidates. The final loss we used can be represented as

$$\mathcal{L}_{total} = \lambda_c \mathcal{L}_{conf}^{E} + \lambda_{ep} \mathcal{L}_{pos}^{E} + \lambda_{dp} \mathcal{L}_{pos}^{D} + \lambda_r \mathcal{L}_{rank} + \lambda_s \mathcal{L}_{sort}. \tag{8}$$

Since the ranking supervision depends on the quality of the candidate line segments from the encoder, poor prediction results at the beginning will affect the training of the decoder. Therefore, we warm up the model for several epochs. Only the encoder is supervised during the beginning of training. Then both encoder and decoder are jointly trained after the encoder can predict meaningful line segments.

| Method | Wireframe | | | | | YUD | | | | | FPS |
|---|---|---|---|---|---|---|---|---|---|---|---|
| | $sAP^5$ | $sAP^{10}$ | $sF^{10}$ | $sF^{15}$ | LAP | $sAP^5$ | $sAP^{10}$ | $sF^{10}$ | $sF^{15}$ | LAP | |
| LSD [32] | 6.7 | 8.8 | - | - | 18.7 | 7.5 | 9.2 | - | - | 16.1 | 100.0 |
| DWP [13] | 3.7 | 5.1 | - | - | 6.6 | 2.8 | 2.6 | - | - | 3.1 | 2.2 |
| AFM [35] | 18.3 | 23.9 | - | - | 36.7 | 7.0 | 9.1 | - | - | 17.5 | 14.1 |
| LGNN [23] | - | 62.3 | - | - | - | - | - | - | - | - | 15.8 |
| TP-LSD [14] | 57.6 | 57.2 | - | - | 61.3 | 27.6 | 27.7 | - | - | 34.3 | 20.0 |
| L-CNN [45] | 58.9 | 62.8 | 61.3 | 62.4 | 59.8 | 25.9 | 28.2 | 36.9 | 37.8 | 32.0 | 16.6 |
| M-LSD [8] | 56.4 | 62.1 | - | - | 61.5 | 24.6 | 27.3 | - | - | 30.7 | 115.4* |
| M-LSD†[8] | 63.3 | 67.1 | - | - | 64.2 | 27.5 | 28.5 | - | - | 32.4 | 32.9 |
| HAWPv2 [38] | 65.5 | 69.5 | 66.4 | 67.4 | - | 28.2 | 30.4 | 41.0 | 42.0 | - | 45 |
| LETR [33] | 59.2 | 65.6 | 66.1 | 67.4 | 65.1 | 24.0 | 27.6 | 39.6 | 41.1 | 32.5 | 5.4 |
| RANK-LETR [31] | 65.0 | 69.7 | 66.7 | 67.7 | 65.6 | 27.6 | 30.1 | 39.7 | 40.6 | 34.1 | 9.0 |
| RANK++LETR (Ours) | **67.9** | **72.1** | **68.8** | **69.7** | **68.3** | **28.8** | **31.2** | **41.2** | **42.1** | **34.8** | 12.4 |

Table 1: Quantitative comparisons on Wireframe [13] and YUD [6] datasets. We compare our proposed method with LSD [32], DWP [13], AFM [35], LGNN [23], TP-LSD[14], L-CNN [45], HAWPv2 [37], M-LSD [8], LETR [33] and RANK-LETR [31] methods. M-LSD† denotes the approach of combining M-LSD and HAWP. Average precision (sAP), F-score measurement (sF) and line matching average precision (LAP) are used as metrics for comprehensive comparisons. Our method outperforms previous SOTA methods in prediction accuracy and gets faster inferring speed than other Transformer-based methods.

## 4 Experimental Results

### 4.1 Experimental Setup

#### 4.1.1 Datasets ans Metrics

We conduct our experiments in two publicly available datasets including the Wireframe dataset [13] and the YorkUrban dataset [6], which are widely used as line segment detection benchmarks. The Wireframe dataset contains 5,000 training and 462 testing images of man-made environments, while the YorkUrban dataset contains 102 testing images. The model is only trained on the Wireframe dataset and tested on both Wireframe and YorkUrban datasets as a typical protocol [14; 45]. For comprehensive comparison, we evaluate our models based on average precision (sAP), F-score measurement (sF) and line matching average precision (LAP). For fair comparison, we select no more than 500 prediction lines with high confidence scores of each image for quantitative analysis.

#### 4.1.2 Implementation Details

Our training and evaluation are implemented in PyTorch. We use 4 NVIDIA V100 GPUs for training and 1 GPU for evaluation. We train our model for 240 epochs for warming up and 120 epochs for jointly optimizing. The learning rate is set as $5 \times 10^{-4}$. The image size and the batch size are set as $512 \times 512$ and 8, respectively. We use the AdamW optimizer and set weight decay as $10^{-4}$.

The results of our method are predicted on the features of the resolution of $128 \times 128$. $\lambda_c$, $\lambda_{ep}$, $\lambda_{dp}$, $\lambda_r$, $\lambda_s$ are set to $1, 10, 10, 1, 1$, respectively. Moreover, we use auxiliary loss on the early layer in the Transformer-based encoder with a factor of 0.8. $K$ is set to 4 for sampling. Up to 500 line segments with high scores are detected with NMS for comparison in our method.

### 4.2 Comparison with the SOTA

We compare our method with previous state-of-the-art methods including LSD [32], DWP [13], AFM [35], LGNN [23], TP-LSD [14], L-CNN [45], HAWPv2 [37], M-LSD [8], LETR [33] and RANK-LETR [31]. All the methods are learning-based methods except the classical LSD. M-LSD†denotes the method of combining M-LSD and HAWP. LETR, RANK-LETR and our proposed RANK++LETR take Transformer as key architecture while other approaches mainly use convolutional neural networks. Some methods such as [26; 25] are not chosen in the comparison because they are designed to tend to generate finer line segments. Thus, it is unfair for these methods to compare on

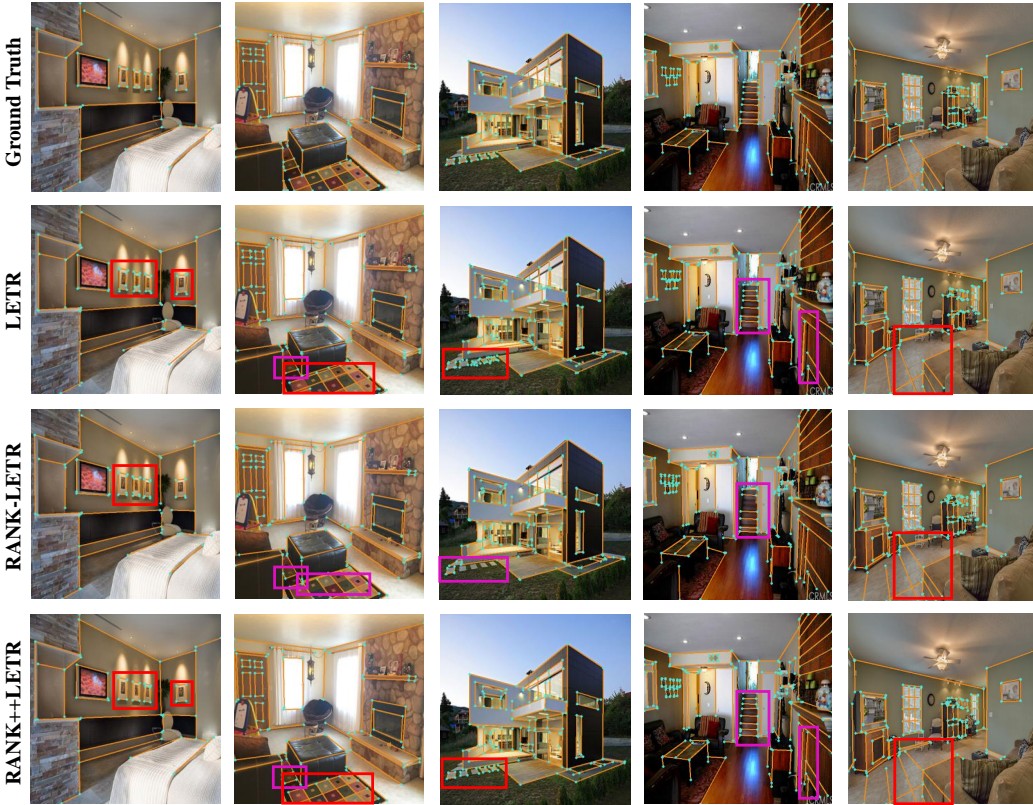

Figure 3: Visual examples of line segment detection results of Transformer-based methods including LETR [33], RANK-LETR [31] and our proposed RANK++LETR on the Wireframe dataset [13]. For a better visual experience, we highlight some significant differences of accurate detection with red bounding boxes and complete detection with purple bounding boxes. Our method tends to produce more accurate and complete line detection results.

standard wireframe and YorkUrban Dataset. The comparisons are conducted on the Wireframe dataset [13] and the YorkUrban dataset [6] and the results are listed in Table 1. The proposed RANK++LETR outperforms all previous SOTA methods in prediction accuracy. Especially, RANK++LETR gets about **2.9** percents improvement over RANK-LETR and **2.4** percents improvement over HAWPv2 on $sAP^5$ metric.

Moreover, RANK++LETR demonstrates superior efficiency compared to other Transformer-based approaches, primarily attributed to its minimalist architectural design that eliminates computationally intensive components such as global attention mechanisms and rotation augmentation operations. It is worth mentioning that a speed gap still persists when benchmarked against optimized CNN-based implementations, where we think that it is mainly due to the inherent computational complexity of Transformer architectures versus the convolutions. Our future work will focus on exploring lightweight Transformer or CNN-guided architecture for more efficient line detection approaches.

Visual examples of line segment detection results of Transformer-based methods including LETR, RANK-LETR and our proposed RANK++LETR on the Wireframe dataset are shown in Fig. 3. Our method tends to produce more accurate and complete line detection results. We highlight some significant differences in accurate detection with red bounding boxes and complete detection with purple bounding boxes.

To explore the generalization capability of the proposed method in non-structured scenarios, we directly applied the trained model to the NKL dataset [44] for semantic line detection. This dataset primarily contains natural landscape images. The detection results are shown in the Fig.4. The experiment demonstrates that even without training on the NKL dataset [44], our model still exhibits perception ability for semantic lines.

| Line Optimizer | LADA | Ranking Loss | w/o Warm-up | R:S | Attention Points | Wireframe sAP$^5$ | sAP$^{10}$ |
|---|---|---|---|---|---|---|---|
| - | - | - | - | - | - | 63.2 (↓ 2.3) | 68.1 (↓ 2.7) |
| ✓ | - | - | - | - | 4 | 65.5 (-) | 70.8 (-) |
| ✓ | ✓ | - | - | - | 4 | 67.0 (↑ 1.5) | 71.4 (↑ 0.6) |
| ✓ | - | ✓ | - | 1:1 | 4 | 67.2 (↑ 1.7) | 71.4 (↑ 0.6) |
| ✓ | ✓ | ✓ | - | 1:1 | 4 | **67.9** (↑ 2.4) | 72.1 (↑ 1.3) |
| ✓ | ✓ | ✓ | ✓ | 1:1 | 4 | 66.0 | 70.6 |
| ✓ | ✓ | ✓ | - | 1:0 | 4 | 67.3 | 71.6 |
| ✓ | ✓ | ✓ | - | 0:1 | 4 | 48.7 | 51.4 |
| ✓ | ✓ | ✓ | - | 1:1 | 2 | 67.4 | 71.5 |
| ✓ | ✓ | ✓ | - | 1:1 | 8 | **67.9** | **72.2** |

Table 2: Ablation and parameter study of our method on the Wireframe [13] dataset. We first construct a baseline method without decoder according to our modeling and then gradually add different components to explore their relevance and impact. Experimental results show that a second optimization can bring performance improvement, and both the LADA module and ranking loss contribute to better detecting results. Different numbers of attention points and weights of RS losses are also tested for a comprehensive study.

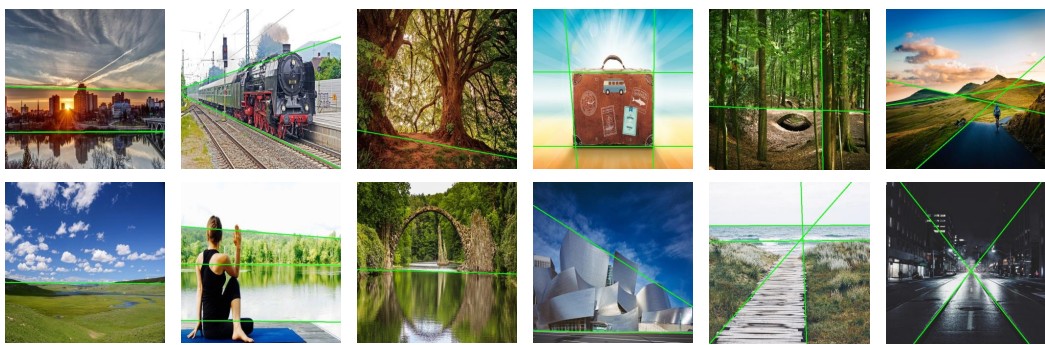

Figure 4: Visual examples for non-structured environments on NKL dataset [44]. Our method demonstrates strong generalization capability in semantic line detection scenarios.

## 4.3 Ablation and Parameter Study

To verify the effectiveness of components and find the influence of the hyperparameters in our proposed approach, we conduct an ablation and parameter study of RANK++LETR. The experience is conducted on the Wireframe dataset and *sAP* results are reported in Table 2.

As a baseline method, we apply ResNet50 as feature extractor and 6 layers Transformer encoder from classical deformable DETR for line segment detection with a matched prediction strategy, where the feature point closest to the centroid of a line segment is responsible for predicting it. Different from RANK-LETR, branch network and rotation enhancement network are no longer used. Based on the baseline, we use 6 layers deformable Transformer decoder and construct the complete proposal generating and candidate optimizing pipeline, where the selected proposals are used to initialize the inputs of the decoder. It reveals a significant performance gap between the proposed pipeline and baseline framework, which proves the effectiveness of our detection modeling. The line-aware deformable attention and ranking-based loss are then added individually. Both of them can bring obvious performance improvement. By combining the LADA module and ranking-based Loss together, we verify that our proposed method can get the best results benefiting from each novel component. Moreover, we find the warmup of encoder is also important to get a faster convergence for better results. The experimental results are listed in Table 2. The contribution of each component can be found during the performance differences.

The parameter study is conducted on LADA module and ranking-based losses. For LADA module, we explore the influence of the number of attention points, by changing the default 4 to 2 and 8, respectively. We find performance is not sensitive to the number of attention points when it exceeds

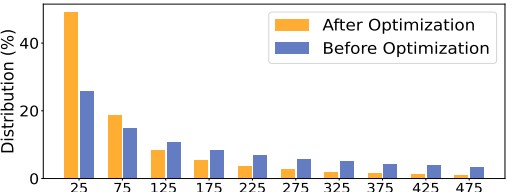

Figure 5: Statistics of the distribution on the rank of corresponding ground truth being recalled. The ranking-based supervision contributes to gaining high ranking distribution of the prediction, yielding a better line segment detection performance.

Table 3: Evaluation of the outputs before and after optimization, respectively. Adding line optimizer for learnable ranking the confidence scores and refining the positions brings an intuitive detecting performance improvement.

| Metric | Before Optimize | After Optimize |
|--------|-----------------|----------------|
| $sAP^5$ | 64.1 | **67.9** |
| $sAP^{10}$ | 68.8 | **72.1** |
| $sF^5$ | 64.5 | **66.3** |
| $sF^{10}$ | 67.4 | **68.8** |

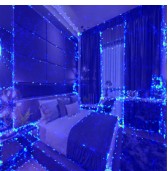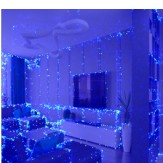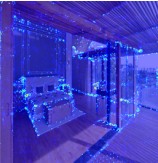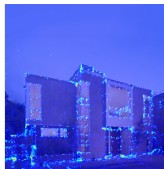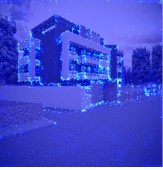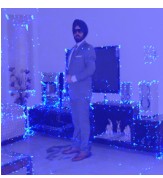

Figure 6: Attention maps of our proposed line-aware deformable attention module (LADA) in the decoder. We show the attention map of the last decoder layer for clear viewing. Brighter areas mean more attention. The attention points generally appear around the image lines, making the proposed LADA easier to capture line features.

4, which may be because it obtains enough useful information such as endpoints and edges. For ranking-based supervision, we test the ranking loss and sorting loss individually to find their roles in training. We observe that only adding ranking loss can bring limited improvement and only using sorting loss will even get worse performance. We think both loss terms should be used together in the task, which may be because they complement each other.

## 4.4 Analysis and Interpretation

In order to gain a deeper understanding of the proposed method, we conduct further analysis and feature visualization to exhibit intermediate process. As our method uses encoder to get initial proposals and optimize the score and position with decoder in successive processing, an intuitive way to explain the pipeline is to compare the predicting performance of the two outputs in one model directly. As demonstrated in Table.3, the line optimizer brings an intuitive detecting performance improvement.

We then compare the ranking quality to verify the effectiveness of the supervision and further explore the underlying reason for performance improvement. For the output results before and after line optimization, we select 500 line segments with high confidence and gather them into two groups, respectively. For each ground truth line segment, the closest line segment can be found from each group and the ranking of the line segment among its group can be recorded. It indicates that the corresponding ground truth will be recalled at the ranking in these predictions. It is obvious that a better method should gain higher ranking quality. In other words, with fewer predictions, more correct line segments can be detected. We statistically analyze the distribution of the rank of corresponding ground truth being recalled and the results are shown in Fig.5. The ranking-based supervision contributes to gaining high ranking distribution of the prediction, yielding a better line segment detection performance.

We also visualize the distribution of attention points in line-aware deformable attention (LADA). 6 images are randomly selected with the attention heat map masking on, which is shown in Fig.6. We show the attention map of the last decoder layer for clear viewing. Brighter areas mean more attention. The attention points generally appear around the image lines, demonstrating that the proposed LADA is easier to capture line features and more suitable for line detection tasks.

In addition to the module innovation, from a holistic perspective, the proposed method leverages the encoder-decoder design where the encoder generates line segment proposals while the decoder

performs confidence re-ranking and position refinement of these segments. For the encoder, our method relies on its recall capability, whereas the decoder provides a secondary opportunity to enhance detection performance by ranking confidence and optimizing locations for the recalled line segments. Theoretically, since the encoder itself is trained as a line segment detector, cases where the proposed encoder itself demonstrates significantly inferior recall performance compared to other detection methods are unlikely to occur.

## 5 Conclusion

In this work, we develop proposal-based line segment detection methods with a novel pipeline where the high-quality line proposals are ranked and optimized with learnable features. To achieve this goal, we specially design a novel line-aware deformable attention (LADA) module in which attention positions are distributed in a long narrow area and can align well with the elongated geometry of line segments. For better supervising the ranking of selected proposals, ranking-based losses are employed and modified with proper contiguous labels generation to adapt line segment detection task. Building on the above techniques, we construct a novel line segment detection model with encoder-decoder architecture named RANK++LETR. Extensive experiments show that our method outperforms previous SOTA methods in prediction accuracy and gets faster inferring speed than other Transformer-based methods.

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
