# OpenReview forum: "RANK++LETR: Learn to Rank and Optimize Candidates for Line Segment Detection"
_NeurIPS.cc/2025/Conference — NeurIPS 2025 poster_

### Official Review · Reviewer_2ecv · 2025-06-11

**Clarity:** 2
**Significance:** 2
**Originality:** 2
**Rating:** 2
**Confidence:** 5

**Summary:**

This paper introduces a proposal-based line segment detection method with a pipeline where the high-quality line proposals are ranked and optimized with learnable features. To achieve this goal, a line-aware deformable attention (LADA) module is proposed, where attention
positions are distributed in a long narrow area and can align well with the elongated geometry of line segments. For better supervising the ranking of the selected proposals, ranking-based losses are employed and modified with proper contiguous labels generation to adapt line segment detection task.

**Questions:**

(1) To explain in theory and experimentally how the proposed attention module literally helps the detection of line segments.
(2) To explain in theory and experimentally how the proposed ranking mechanism literally helps the detection of line segments.
(3) To show the weaknesses of the proposed solution.

**Ethical Concerns:**

["NO or VERY MINOR ethics concerns only"]

**Final Justification:**

I have read the rebuttals of the authors but not convinced by the arguments. I would not change my score previously entered.

**Limitations:**

Not really. It would be beneficial to discuss when the proposed method fails in the experiments.

**Paper Formatting Concerns:**

no formatting concern.

**Quality:**

2

**Strengths And Weaknesses:**

Quality -

The paper has been well structured, with good balance between theory and experiments. The results seem to suggest that the proposed approach works better than the other state of the arts. However, the connection between the attention module and the line segmentation is not clear in the current form. It is not clear how the ranking of line candidates can contribute towards the optimized results.

Clarity -

The major concern is that it is not clear how the attention module makes the samples better aligned with semantical image features. In addition, it is not clear how the ranking theory improves the alignment of line segments. There is no evidence in the theory and experiments for justifying these two aspects.

Significance -

The topic is important. The proposed solution seems to be incremental, compared against the other methods. The results also look minor improvement.

Originality -

The proposed method is not original. In fact, it is driven by the applications where several established modules are combined for better line segment detection.

---

> ### Author Rebuttal · Authors · 2025-07-31
>
> Thank you for your detailed review and valuable suggestions. We will try our best to clearly explain your concerns point by point.
>
> Q1: Evidence in the theory and experiments for justifying LADA and Line Ranking Loss. Connection/Contribution are not clear.
> A1: (1) Connection and theory of LADA are in Sec 3.2. We design line-aware deformable attention (LADA) module in which attention positions are distributed in a long narrow area and can align well with the elongated geometry of line segments. Theoretical expression style is similar to classical works such as Deformable-DETR([46]).
> (2) Connection and theory of Line Ranking Loss are in Sec 3.3. We propose line ranking loss in line segment detection task with the design of contiguous labels according to the detection quality. Theoretical expression style is similar to classical works such as RSLoss ([24]), RANKED ([4]).
> (3) The effectiveness of our proposed LADA and Line Ranking Loss are shown in our ablation study in Sec 4.3, which bring 1.5 and 1.7 improvement respectively on the $sAP_5$ metric.
> (4) Visualization of LADA is exhibited in Fig.5. Ranking quality can be reflected in Fig.4.
>
> Q2: Concern about the innovation.
> A2: Our innovation can be summarized as: (1) propose a pure end-to-end learnable propose-optimize line segment detection framework. Moreover, we consider that introducing and modifying proper module structures appropriately to adapt to a specific task (LSD) for improving performance/ solving problems is also an innovation, just like LETR itself. (2) design line-aware deformable attention (LADA) for line segment detection. (3) line ranking loss. The key innovation here is finding ranking-based loss naturally suitable for the optimizing process in our pipeline, and designing the contiguous labels $l_i$ for the LSD task. (4) Performance: more accurate than previous methods and faster than previous Transformer-based method.
>
> Q3: Concern about the improvement.
> A3: We argue that our method gets an obvious improvement. Our method get 2.4/2.6 improvement on $sAP_5$, $sAP_{10}$ compared with previous the SOTA method, and get 2.9/2.4 improvement compared with the SOTA Transformer-based method. These two previous SOTA method achieve 1.4/0.8 and 1.7/2.6 improvement on the same dataset, respectively.
>
> Q4: Show the weaknesses of the proposed solution.
> A4: The main weakness is the speed gap benchmarked against optimized CNN-based implementations, which is mentioned at line221-227 in the original paper.

---

> > ### Comment · Reviewer_2ecv · 2025-08-07
> >
> > The authors have addressed some of the concerns raised by the reviewer, in particular, experimental justifications. However, the authors responded to the concerns about the theoretical justifications weakly. For example, they stated 'Theoretical expression style is similar to classical works such as RSLoss ([24]), RANKED ([4])' - if this is true, where is the novelty of the proposed solution?

---

> > > ### Author Response · Authors · 2025-08-07
> > >
> > > Thank you for your detailed comments. We try to explain your concerns as follows:
> > >
> > > Q1: “Theoretical expression style is similar to classical works such as RSLoss ([24]), RANKED ([4])” means poor novelty.
> > >
> > > A1: Please note that the term "theoretical expression **style**" refers specifically to **the way to present our theoretical expression** for our ranking loss (not the theoretical expression itself). It is similar to classical works such as RSLoss ([24]), RANKED ([4]) in aspects such as:‌
> > >
> > > (1) Similar aspects we are concerned about: Introducing the definitions of Rank/Sort functions and the construction of the continuous function;
> > > (2) Similar mathematical notation and formula formatting: We intentionally adopted this style to facilitate readers familiar with those works in better following our approach.
> > >
> > > ‌However, this does not mean the same novelty or contribution as the previous works.‌ Our core innovations in ranking loss are:
> > > (1) ‌First introduction of Rank/Sort functions to LSD‌, driven by the key theoretical insight that confidence ranking of line segments critically impacts detection accuracy;
> > > (2) ‌Novel construction of the LSD-specific continuous function $l_i$‌, enabling the formal definition of Rank/Sort objectives in this task.
> > > ‌These contributions are unprecedented and distinct from all existing methods.
> > >
> > > Q2: Theoretical justifications/novelty
> > >
> > > A2: Besides the SOTA performance shown in the experimental results, three novel pipeline or modules are presented in our work.
> > >
> > > (1) We use a Transformer-based encoder-decoder architecture to build an end-to-end learnable propose-optimize line segment detection framework. The running process is described in section 3.1.
> > >
> > > (2) We design LADA module where attention positions are distributed in a long narrow area and can align well with the elongated geometry of line segments. The formula derivations of LADA start from a general attention module (Eq.(1)) with the whole attention grid (Eq.(2)). Then we show the learnable selective attention points around a reference point in deformable attention (Eq.(3)).  Finally, we introduce our LADA by defining the reference points sampled along the line segments for better perceiving line features (Eq.(4)). The theoretical justifications are organized step by step in section 3.2.
> > >
> > > (3) We design line ranking loss to LSD‌, driven by the key theoretical insight that the confidence ranking of line segments critically impacts detection accuracy. The formula derivations start by constructing the contiguous label function $l_i \in [0,1]$ (Eq.(5)) to describe the line detection performance based on the location. Then we introduce the ranking loss and sorting loss to rank the positives over negatives (Eq.(6)) and sort the positives w.r.t. prediction quality (Eq.(7)) based on the contiguous label we constructed. The theoretical justifications are organized step by step in section 3.3.
> > >
> > > ‌Moreover, some related works listed in the rebuttal can also be suggested as references to objectively evaluate our theoretical justifications.
> > >
> > > We hope the above explanation can solve your concern. Any further questions are kindly welcome.

---

> > > > ### Comment · Area_Chair_zcP2 · 2025-08-08
> > > >
> > > > Dear Reviewer 2ecv,
> > > >
> > > > The authors have responded to your concern regarding experimental justifications. Could you please review their reply and let us know whether it addresses your concern?
> > > >
> > > > Best regards
> > > > Your AC

---

> ### Author Response · Authors · 2025-08-06
>
> We are grateful for your clear and thoughtful reviews. We are willing to continue addressing any uncertainties and improving our work if further questions or suggestions are shared.

---

### Official Review · Reviewer_NZBC · 2025-06-18

**Clarity:** 3
**Significance:** 2
**Originality:** 2
**Rating:** 4
**Confidence:** 4

**Summary:**

This paper presents RANK++LETR, a Transformer-based method for line segment detection that introduces two key innovations: (1) a line-aware deformable attention (LADA) module to better capture elongated line features by aligning attention points along candidate segments, and (2) a ranking-based supervision strategy that optimizes confidence scores by learning to prioritize high-quality predictions. Building on deformable DETR, the model separates line proposal generation (encoder) from refinement and re-ranking (decoder), achieving state-of-the-art accuracy on Wireframe and YorkUrban datasets while maintaining efficient inference. The work advances structured scene understanding by improving both detection quality and geometric alignment of line segments.

**Questions:**

- Most of the scenarios shown in the paper are those with more structured features. Can the algorithm perform better in some scenarios with more unstructured features, or has it been tested on relevant data? As far as the dataset is concerned, the dataset used in this paper does not seem to be a particularly large dataset (5000 training images).
- The lines are all 3D structures, can 2D inspection have a large application?

**Ethical Concerns:**

["NO or VERY MINOR ethics concerns only"]

**Final Justification:**

Thanks to the author for the detailed response. I think the author's work is valuable, but given some of the existing problems with the article, I keep my original rating for the weak acceptance.

**Limitations:**

The innovativeness of this paper is limited overall, although the results of this paper are particularly good. In addition, the FPS of the method in this paper is relatively low, and some training and inference overheads are not reported.

**Paper Formatting Concerns:**

None.

**Quality:**

3

**Strengths And Weaknesses:**

Strengths:
- This paper introduces an innovative line-aware deformable attention (LADA) module that effectively addresses the geometric mismatch in standard attention mechanisms by distributing attention along elongated regions, significantly improving feature alignment for linear structures.
- The experimental validation demonstrates state-of-the-art performance on standard benchmarks (Wireframe and YorkUrban datasets), with particular emphasis on the method's computational efficiency, achieving faster inference speeds than comparable Transformer-based approaches while maintaining high precision. The clear ablation studies and visualizations further strengthen the technical rigor of this work.

Weaknesses:
- Limited innovation. The whole Transformer structure is more similar to LETR. The introduction of the ranking-based network layer and ranking-based loss function in the Transformer is also existing research.
- While faster than Transformer baselines, it’s still slower than optimized CNNs (e.g., M-LSD). No FLOPs/parameter comparison provided. Multi-scale encoder-decoder imposes memory overhead (no graphics memory footprint reported)
- Key thresholds (e.g., distance factor δₗ in Eq. 5) lack theoretical justification.

---

> ### Author Rebuttal · Authors · 2025-07-30
>
> Thank you for your detailed review and valuable suggestions. We will try our best to clearly explain your concerns point by point.
>
> Q1: Concern about innovation
> A1: Our innovation can be summarized as: (1) propose a pure end-to-end learnable propose-optimize line segment detection framework. Moreover, we consider that introducing and modifying proper module structures appropriately to adapt to a specific task (LSD) for improving performance/ solving problems is also an innovation, just like LETR itself. (2) design line-aware deformable attention (LADA) for line segment detection. (3) line ranking loss. The key innovation here is finding ranking-based loss naturally suitable for the optimizing process in our pipeline, and designing the contiguous labels $l_i$ for the LSD task. (4) Performance: more accurate than previous methods and faster than previous Transformer-based method.
>
> Q2: Slower than optimized CNNs/FLOPs/parameter
> A2: We consider it a drawback of our approach, as discussed in our original paper. CNNs are truly fast. Our future work will employ efficient Transformers or even replace some modules with CNNs. We keep trying our best to establish a new milestone in LSD. Comparison on FLOPs and parameter are listed below. Rotation augmentation operation imposes much calculation overhead for Rank-LETR.
>
> Metric|Rank-LETR|Rank++LETR
> --------|-----|------
> GFLOPs |138|301
> parameter| 27M|25M
>
>
> Q3: Theoretical justification for selecting $\delta_l$:
> A3: We choose distance factor $\delta_l$ according to existing metrics AP/sF_x, for x in [5, 10, 15]. Thus $\delta_l$ can be chosen as [1/5, 1/10, 1/15]. We choose it 1/10 in our experiments.
>
> Q4: Performance on unstructured scenes:
> A4: For non-structured scenes, we directly applied the trained model to the semantic line detection dataset (NKL). We find that the detection of the boundary between linear strip-shaped traces is more in line with human perception (such as sea level, bridge, road, grass), while the detection of the boundary between serrated traces is relatively random (mountain-sky boundary, rugged tree crowns). We guess that the reason for this is the large gap between the line patterns in such scenes and the images in the training set. Since no figures can be supplemented in the rebuttal, we will add some visualization of non-structural scenes in our final version.
>
> Q5: Application in 2D inspection:
> A5: Yes, it can be directly used in 2D inspection, such as document upright adjustment, graphic vectorization.

---

> > ### Author Response · Authors · 2025-08-03
> >
> > Sorry for reversing the results in the table. The corrected table should be
> >
> > Metric|Rank++LETR|Rank-LETR
> > --------|-----|------
> > GFLOPs |138|301
> > parameter| 27M|25M

---

> > ### Comment · Reviewer_NZBC · 2025-08-04
> >
> > Thanks to the author for the detailed response. I think the author's work is valuable, but given some of the existing problems with the article, I keep my original rating for the weak acceptance.

---

> > > ### Author Response · Authors · 2025-08-06
> > >
> > > Thank you again for your detailed review and valuable suggestion. We are glad to see that our work is regarded as valuable. We are also willing to continue addressing any uncertainties and improving our work if further questions or suggestions are shared.

---

### Official Review · Reviewer_XaiL · 2025-06-29

**Clarity:** 3
**Significance:** 2
**Originality:** 3
**Rating:** 5
**Confidence:** 3

**Summary:**

This paper addresses the task of line segment detection. The main claim is that previous transformer-based methods assign detection scores that do not reflect the actual quality of the predicted lines.

The proposed method builds on an encoder-decoder transformer architecture with deformable attention. The encoder makes an initial set of line predictions. The novelty lies in the decoder, which refines these predictions and adjusts the scores to better align with their geometric quality.

To enforce this alignment by the decoder, the method adopts ranking-based supervision, similar to RANK-LETR.
A new deformable attention module is proposed, which samples reference points along narrow regions.

The method outperforms prior state-of-the-art approaches. Ablation studies highlight the benefits of using a decoder stage, in contrast to encoder-only designs like RANK-LETR.

**Questions:**

1. Clarify the difference and the novelty between your proposed deformable attention and the one of the decoder of DINO-DETR. (see my point in weakness)
2. Would using DINO-DETR directly, combined with the proposed loss functions of this paper, yield similar results? This possibility should be addressed more clearly.
3. Can you show examples of failure cases and explain them? For example, missed detections or cases where predicted scores and line quality are still not well aligned.

**Ethical Concerns:**

["NO or VERY MINOR ethics concerns only"]

**Final Justification:**

The authors clearly answer to my main concerns by provided a detailed rebuttal.

**Limitations:**

yes

**Quality:**

3

**Strengths And Weaknesses:**

Strengths:
Writing: The paper is well written, easy to follow, and clearly presented.
Related Work: To my knowledge, the related work is satisfactory.
Technical Contribution: Using a decoder to align predicted scores with line quality is a novel and interesting idea. It makes good use of the encoder-decoder architecture, in contrast to RANK-LETR which uses only an encoder.
Experiments: The method is compared to strong SOTA baselines and outperforms them. Ablation studies are provided, showing the benefits of using a decoder to optimizer the prediction, the impact of the number of attention points and different loss components used to train the model.


Weakness
I am really not convinced by the proposed deformable attention module. It seems very similar to the decoder deformable attention in DINO-DETR, which uses bounding boxes as reference points. If these boxes collapse onto a line, the behavior appears equivalent to the proposed method. Please clarify the difference and the novelty.

Compared to RANK-LETR, the modifications appear relatively small. The main architectural change is the decoder and the deformable attention, which—again—looks very similar to DINO-DETR. Would using DINO-DETR directly, combined with the proposed loss functions of this paper, yield similar results? This possibility should be addressed more clearly.

One of the main claims is improved inference speed. But , according to Table 1, the method is not faster than RANK-LETR or LETR.
The ablation study should include the same proposed  architecture but without the proposed deformable attention module. This should be included to demonstrate that the new module clearly improves performance.

The paper does not discuss limitations or failure cases. Can you show examples of failure cases and explain them? For example, missed detections or cases where predicted scores and line quality are still not well aligned.

Small details:
Please report training in number of iterations, not just epochs. Since an epoch refers to one pass through the dataset, the number of iterations can vary across datasets and is not always clear.

Table 1: Please define FPS. I assume it refers to inference speed, but this should be explicit.

Table 2: It is not immediately clear that the first row corresponds to a baseline without the encoder. Please clarify.

Equation 8: How did you choose the hyperparameter values for the loss weights? Please explain.

Figure 1: Why is there a blue line from the ground truth to the initial position and score boxes? Please clarify its meaning.


Figurte 1.. why is there a blue lin from grounth truth to the initial positon & scores boxes?

---

> ### Author Rebuttal · Authors · 2025-07-30
>
> Thank you for your detailed review and valuable suggestions. We will try our best to clearly explain your concerns point by point.
>
> Q1: Difference between LADA and the attention in DINO
> A1: The difference can be noticed in Fig.2(a). For the attention in DINO, the referencing point is the center point of the candidate box, and the attention points are generally distributed around the referencing point. Please refer to Fig.6 in the Deformable DETR article for a similar visualization. This distribution is not conducive to the perception of narrow targets such as line segments. The LADA uniformly samples the candidate line segments as the referencing points, so that the attention points are distributed along the line segments, which is beneficial for the perception of narrow areas. The results of our ablation study also confirm this point. For attention in models such as DINO, the shape of the candidate box is only related to the boundary of reference points. Not to mention that the collapsed box can only approximate horizontal or vertical line segments.
>
> Q2: Compare with DINO+LSD(+line ranking loss)
> A2: DINO+LSD can be seen as the 2nd row in Table 2 (sAP5:65.5) and the DINO+LSD(+line ranking loss) can be seen as 4th row in Table 2 (sAP5:67.2) in the original paper, which has a performance gap with our proposed method.
>
> Q3: Difference between the proposed method and RANK-LETR
> A3: (1) The proposed method uses a pure learnable Transformer-based decoder to re-rank (scores) and optimize (locations) the line segment simultaneously, while Rank-LETR uses CNN CNN-based manner to re-rank the score with pre-designed hyperparameters. (2) Design line-aware deformable attention (LADA) for line segment detection. (3) Introduce a powerful line ranking loss. The key insight here is finding ranking-based loss naturally suitable for the optimizing process in our pipeline, and designing the contiguous labels $l_i$ for the LSD task.
>
> Q4: Inference speed/Ablation study
> A4: Our method is faster than LETR and Rank-LETR in Table 1. The higher the FPS value is, the faster. In Table 2, we conduct the experiment using a module adding pipeline, and the comparison between with and w/o LADA can be obtained by comparing the results in 2/3 rows and 4/5 rows, respectively. sAP5 increase 1.7 when adding the LADA module compared to the baseline, while decrease 0.7 when subtracting the LADA module. Please note that it also reflects the ability of Line Ranking Loss. The change in improvement is caused by the ability overlapping of the two proposed modules.
>
> Q5: Limitation and failure cases
> A5: The main limitation is the speed gap benchmarked against optimized CNN-based implementations, which is mentioned at line221-227 in the original paper. Most failure cases happen in dense short line segment detecting, e.g., the 3rd column in Fig.3. It is worth noting that this has always been a common problem in learning based methods. Compared to the existing methods, our method has made improvements in this issue.
>
> Q6: Training Iterations
> A6: Our training iteration is set as 225000.
>
> Q7: Explain FPS
> A7: Yes, FPS is short for Frame Per Second, which indicates the number of images the model can process in 1 second. A large value indicates high inference speed.
>
> Q8: Row 1 in the ablation study
> A8: The first row corresponds to a baseline with an encoder and without a decoder.
>
> Q9: Explain the hyperparameter values
> A9: $\lambda_c$, $\lambda_{ep}$, $\lambda_{dp}$ are set as the same as RANK-LETR, $\lambda_r$, $\lambda_s$ are set according to previous study on ranking based losses ([24] in the paper) and parameter study.
>
> R:S|2:1|1:1|1:2
> --------|-----|------|------
> $sAP_5$ | 67.7|67.9|67.1
>
>
> Q10: Explain the blue line from the ground truth to the initial position and score boxes in Fig.1.
> A10: The initial scores and positions are supervised by the ground truth line segments during training.

---

> > ### Comment · Area_Chair_zcP2 · 2025-08-08
> > **provide any feedback on the authors’ response**
> >
> > Dear Reviewer XaiL,
> >
> > Could you provide any feedback on the authors’ response? The AC would appreciate it, as this is a borderline case.
> >
> >
> > Best regards
> >
> > Your AC

---

> > > ### Comment · Reviewer_XaiL · 2025-08-08
> > >
> > > I apologize for the delay for consider your rebuttal due to personal issue. Thank you for the detailed rebuttal. Your responses clearly address my concerns, and I have increased my score accordingly.

---

> > > > ### Author Response · Authors · 2025-08-08
> > > >
> > > > Thank you again for your response and valuable suggestion. We sincerely appreciate your willingness to increase the score.

---

> ### Author Response · Authors · 2025-08-06
>
> We are grateful for your detailed and thoughtful reviews. We are willing to continue addressing any uncertainties and improving our work if further questions or suggestions are shared.

---

### Official Review · Reviewer_3gFX · 2025-07-02

**Clarity:** 3
**Significance:** 4
**Originality:** 2
**Rating:** 5
**Confidence:** 5

**Summary:**

This paper proposes RANK++LETR, an end-to-end Transformer-based framework for line segment detection. The architecture builds upon deformable DETR and introduces a Line-Aware Deformable Attention (LADA) module that aligns attention patterns with elongated line structures. Furthermore, the paper innovates by using ranking-based loss functions for line candidate refinement, improving both scoring and localization of predictions. The method is evaluated on Wireframe and YorkUrban datasets, outperforming prior SOTA in accuracy and inference speed.

**Questions:**

How does RANK++LETR handle complex or cluttered scenes with overlapping or occluded line segments?

Would the model benefit from unsupervised or semi-supervised pretraining to enhance robustness? Can LADA be adapted for use in curved feature detection (arcs or splines)?

What is the impact of increasing the number of attention points beyond 8 on both performance and runtime? How sensitive is the model to inaccuracies in the continuous label generation process?

**Ethical Concerns:**

["NO or VERY MINOR ethics concerns only"]

**Final Justification:**

I am maintaining my overall score of 5 (Accept).

This paper makes a meaningful contribution to line segment detection, combining architectural innovation (LADA) and training strategies (ranking losses) with strong empirical results. The rebuttal further solidifies its clarity and relevance.

**Limitations:**

Yes. The authors adequately acknowledge the limitations of their current dataset scope and explicitly discuss the performance–complexity trade-off.

**Quality:**

3

**Strengths And Weaknesses:**

Strengths:
Solid architectural and methodological enhancements, with clear ablations and improvements over strong baselines.
The LADA module and application of ranking-based supervision for geometric primitives are novel contributions.
Demonstrates strong gains in both accuracy and inference time over previous methods, showing practical viability. Explanations, figures, and experiments are well presented, especially the attention visualizations and ablation study.

Weakness:
Some improvements like ranking losses are incremental over RANK-LETR.
The method still requires multiple components (e.g., LADA, ranking, warmup), which increases implementation complexity.
No discussion on generalizing to open-scene or non-structured environments beyond wireframe-type datasets.

---

> ### Author Rebuttal · Authors · 2025-07-29
>
> Thank you for your detailed review and valuable suggestions. We will try our best to clearly explain your concerns point by point.
>
> Q1: Multiple components increase implementation complexity.
> A1: Ranking loss and warm up are used in the training phase. So they will not affect the efficiency of the model in inference. LADA brings a limited complexity increase, but also an obvious performance improvement.
>
> Q2: Benefit from unsupervised or semi-supervised pre-training?
> A2: It is a very good suggestion. Some studies have tackled the task with unsupervised or semi-supervised approaches (e.g., [25][26] in our original paper). Our future work will focus on unsupervised and semi-supervised methods, such as pre-training the network through bootstrapping or synthetic labeling, as well as possibly performing some appropriate model adaptation. We think that there will be a significant improvement in generalization at least.
>
> Q3: Extend LADA to curved feature detection.
> A3: We strongly believe that LADA can be used for curved feature detection since they almost share the same design motivation, which is to sample the attention points along the high-confidence line/curve features, allowing the model to learn relative information. On the contrary, a traditional attention machine tends to focus around a point.
>
> Q4: Impact of increasing the number of attention points beyond 8.
> A4: We conduct an experiment that increases the number of attention points to 16. The experimental results show that there is not much change in accuracy and inferring time. We think that the accuracy has reached saturation, and an excessive increase in the number of attention points may only continue to increase (not much) computational overhead.
>
> Metric|p=4|p=8|p=16
> --------|-----|------|------
> $sAP_5$ | 67.9|67.9|67.7
> $sAP_{10}$| 72.1|72.2|71.9
>
> Q5: Sensitive to inaccuracies in the continuous label generation.
> A5: We randomly perturb the positions of all candidate lines by 1-5 pixels, which has some impact on $sAP_5$ and little impact on the $sAP_10$. Moreover, the candidate lines are also learned and predicted by the network. So they are inherently inaccurate compared to the ground truth, but generally do not deviate significantly, as analyzed in line282-289 in the original paper.
>
> Q6: Complex scenes including overlapping/ occluded/ non-structured scenes.
> A6: (1) Overlapping： Dealing with overlapping is one of the strengths of our method. Our method recalls complete line segments because of its focus on global information, and will not be segmented or truncated due to overlapping. It is reflected in columns 2/5 of Fig.3 and the result visualization in the supplementary materials (e.g., carpets, windows and cabinets).
> (2) Occluded： For slight foreground occlusion, our method exhibits good anti-occlusion ability, as shown in the 2nd column of Fig.3, and (r1c2), (r1c3), etc., in the result visualization in the supplementary materials (e.g., foreground lights, chairs). For severely occluded scenes, our method considers them as two line segments on both sides of the occlusion, as shown in the supplementary materials (r3c5), which aligns with human intuitive perception.
> (3) Non-structured scenes： For non-structured scenes, we directly applied the trained model to the semantic line detection dataset (NKL). We find that the detection of the boundary between linear strip-shaped traces is more in line with human perception (such as sea level, bridge, road, grass), while the detection of the boundary between serrated traces is relatively random (mountain-sky boundary, rugged tree crowns). We guess that the reason for this is the large gap between the line patterns in such scenes and the images in the training set. Since no figures can be supplemented in the rebuttal, we will add some visualization of non-structural scenes in our final version.

---

> > ### Comment · Reviewer_3gFX · 2025-08-01
> >
> > I thank the authors for their detailed and thoughtful rebuttal. The responses address all my questions effectively and reinforce the paper's technical strengths.
> >
> > Model Complexity and Efficiency
> >
> > The clarification that ranking loss and warm-up only affect training, not inference, is helpful. The minor added complexity from LADA seems justified given the performance gains demonstrated in the ablation study.
> >
> > Generalization, Occlusion, and Scene Diversity
> >
> > The explanations regarding overlapping and occluded line segments are convincing and demonstrate the model's robustness. The attempt to test on the NKL dataset for non-structured scenes is appreciated, though performance in such environments remains an open challenge. I support including additional visualizations in the final version.
> >
> > Design Choices
> >
> > The ablation over the number of attention points (p=4, 8, 16) confirms that the model’s performance saturates and does not significantly benefit from higher attention granularity. The response on label perturbation robustness was also adequate.
> >
> > Future Directions
> >
> > I'm glad to see the authors are considering unsupervised/semi-supervised extensions and the potential adaptation of LADA to curved feature detection. These would be exciting directions for future work.

---

> > > ### Author Response · Authors · 2025-08-03
> > >
> > > Thank you again for your detailed review and valuable suggestion. We are glad to see that our work can be recognized in peer review and have the opportunity to bring inspiration to the community.

---

### Decision · Program_Chairs · 2025-09-17

**Decision:**

Accept (poster)

**Comment:**

This paper proposes a Transformer-based framework for line segment detection. The proposed method is built on deformable DETR and introduces a Line-Aware Deformable Attention (LADA) module that aligns attention with elongated line geometries. It also employs ranking-based supervision to refine candidate proposals and ensure confidence scores reflect prediction quality. Experiments on benchmark datasets show SOTA accuracy with efficient inference.

Four expert reviewers initially raised several concerns including limited novelty, lack of discussion on generalisation to open-scene or non-structured environments, lack of time complexity analysis, and no discussion of limitations or failure cases. The rebuttal addressed most of these concerns, resulting in two accept, one borderline accept and one reject. The main concern from the rejecting reviewer relates to the authors rebuttal statement: "Theoretical expression style is similar to classical works such as RSLoss ([24]) and RANKED ([4])". This phrasing caused confusion for the reviewer. However, the AC believes that the authors provided a clear explanation of the term “theoretical expression style” and thus recommends acceptance.